# The Psychological Reality of Chinese Deliberate Metaphors from the Reception Side: An Experimental Approach

**DOI:** 10.3390/brainsci13020160

**Published:** 2023-01-18

**Authors:** Juanjuan Wang, Jiajun Du, Ting Zheng, Yi Sun

**Affiliations:** 1School of English Teacher Education, Xi’an International Studies University, Xi’an 710128, China; 2Center for Linguistics and Applied Linguistics, Guangdong University of Foreign Studies, Guangzhou 510420, China

**Keywords:** psychological reality, Chinese deliberate metaphors, reception side, experimental approach

## Abstract

While the psychological reality of deliberate metaphors remains in ongoing doubt, this study attests to their psychological reality in the Chinese language. Based upon the definition and main tenets of deliberate metaphor, we propose three hypotheses: Compared with non-deliberate metaphors, addressees pay more attention to deliberate metaphors’ source domains, display a greater tendency to adopt the source domain’s perspective, and spend more time on processing deliberate metaphors. Using Chinese deliberate metaphors as testing materials, we conducted an experiment with 70 Chinese university students whose native language was Chinese. The results confirmed our hypotheses. Compared with non-deliberate metaphors, Chinese deliberate metaphors drew more attention and brought about more perspective changes. Additionally, processing deliberate novel metaphors consumed more time than processing non-deliberate metaphors, thus providing supportive evidence for the psychological reality of Chinese deliberate metaphors. However, less time taken on deliberate conventional metaphors than non-deliberate metaphors might have been caused by our experiment’s task design. In a nutshell, our study statistically proves the psychological reality of Chinese deliberate metaphors from the reception side. Future studies may check our findings with a similar experimental paradigm in other languages.

## 1. Introduction

With a fruitful 40-year history [1,2], Conceptual Metaphor Theory holds the main tenet of conceptual mapping from the source domain to the target domain [3] (p. 7). When processing a metaphorical expression, people experience cross-domain mapping from the original domain to the receptive domain (comparison) in their minds. Nevertheless, metaphors might be processed metaphorically (by comparison) or just as forms of categorization [4] (p. 363). That is to say, metaphors may not be processed metaphorically—a concept termed “the paradox of conceptual metaphor” [5] (p. 221). To resolve this paradox, Steen [6] proposed the concept of deliberate metaphor, defined as a metaphor purposely used to attract addressees’ attention and change their perspective on a topic or referent [5] (p. 222). Whether the linguistic metaphor would be processed metaphorically in thought was determined depending on whether it was used deliberately in communication [5] (p. 222). In this way, deliberate metaphor is the metaphor processed metaphorically, and the paradox is thus resolved.

Accordingly, deliberate metaphor adds a third dimension to metaphor study, the communication dimension [5]. It invites addressees to view the topic from the perspective of the source domain. This function makes it a valuable tool for communicating knowledge in academic discourse, science communication, and other communication of complex phenomena [7]. In these communications, addressers deliberately use metaphors to inform or explain some difficult concepts or topics, which finally change the addressees’ perspectives [5].

However, the concept of deliberate metaphor has received severe criticism. On the one hand, because knowing metaphor producers’ deliberateness from outside their minds is impossible, ensuring whether they use the metaphor deliberately or not is also impossible [8]. On the other hand, no empirical evidence has been provided to testify to receivers’ conceptual mappings during deliberate metaphor processing. Gibbs [9] once attempted to verify receivers’ perspective change in the presence of pragmatic signals (also called tuning devices, including certain discourse markers, intensifiers, and comparatives and the like, such as ‘well’, ‘quite’, and ‘like’) from deliberate metaphors. Results failed to support that assumption. These criticisms led to doubt on the psychological reality of deliberate metaphor [10]. To further its development, then, deliberate metaphor needs supportive empirical evidence [11].

Therefore, this study attests to the psychological reality of Chinese deliberate metaphors from the reception side. Essentially, “when a speaker or writer uses a metaphor deliberately, that is, as a metaphor in order to make the addressee deliberately understand one thing in terms of something else, the sender forces the addressee to attend to the source domain as a domain that lies outside the current domain of discourse, and to view the target domain from that perspective” [12,13,14,15,16] (p. 54). This explanation depicts two aspects when the addressee understands a deliberate metaphor: attention to the source domain and adoption of the source domain’s perspective (indicated by underlining). Based on the definition and major tenets of deliberate metaphor, we propose two hypotheses about the reception process: Compared with non-deliberate metaphors, receivers pay more attention to deliberate metaphors and display a greater tendency to adopt their perspectives. At the same time, since response time can be taken as an index of the mental process because it decodes mental operation stages [17] (p. 152), we also hypothesize that receivers spend more time processing deliberate metaphors. Experimental results confirmed our hypotheses, verifying the psychological reality of Chinese deliberate metaphors from the reception side.

## 2. Related Studies

### 2.1. The Paradox of Conceptual Metaphor Theory

The deliberate metaphor was put forward in response to the paradox in conceptual metaphor processing. Conceptual Metaphor Theory came into view with the publication of *Metaphors We Live By*, coauthored by Lakoff and Johnson [3], and it was further developed in their subsequent individual works [18,19,20]. The first main tenet of Conceptual Metaphor Theory is the ubiquity of metaphor, in that “metaphor is pervasive in everyday life, not just in language but in thought and action” [3] (p. 3). Our everyday language, conversations, and even behaviors are metaphorical in nature [20]. The second well-known tenet of Conceptual Metaphor Theory is conceptual metaphors as suggested by the term itself. “Metaphorical expressions in our language are tied to metaphorical concepts in a systematic way” [3] (p. 7). That is, metaphorical expressions in language originate from metaphorical structures in thought. Generally, a metaphor employs a relatively more concrete concept (the source domain) to understand a relatively more abstract concept (the target domain) [20]. Therefore, the linguistic expression “He *attacked* every weak point in my argument” activates the conceptual metaphor “ARGUMENT IS WAR” in people’s minds [3].

Nevertheless, these tenets have encountered attacks in subsequent studies. Large-scale analysis showed that only 14% of corpus expressions were metaphorical [21]. This raised a severe challenge to the ubiquity of conceptual metaphors [7]. If only a small sector of language is metaphorical, metaphors are not necessarily what we live by [2]. Moreover, researchers have doubted whether conceptual metaphors really influence our use and understanding of linguistic metaphors [8]. Several experimental studies concluded that linguistic metaphor processing did not actually involve cross-domain conceptual mapping [22,23]. In this way, the model of conceptual metaphors may not match our cognitive reality [7]. Conceptual metaphors are not necessarily processed in a metaphorical manner, and that forms the paradox of conceptual metaphor [5].

### 2.2. Three Rounds of Debates on Deliberate Metaphor

In view of this paradox, Steen advanced the term “deliberate metaphor” [5,6], defined as “a deliberate invitation for the addressee to adopt a different perspective [the perspective intended by the addresser of the deliberate metaphor]” [5] (p. 222). A metaphor is suggested to be deliberate when it is apparently intended to remodel the addressee’s perspective in accordance with the employed source domain. In this way, deliberate metaphors are understood in a metaphorical manner. This proposal initiated three rounds of debate—in 2011, 2015, and 2017.

The initial round [8,12,13,24,25,26] mainly focused on a theoretical discussion of deliberateness from different perspectives: How can we justify producers’ deliberateness in metaphor production? How can we verify receivers’ attention to deliberate metaphors and subsequent cross-domain mappings? Does deliberateness equate to consciousness? What is the relationship between deliberateness, conventionality, consciousness, and awareness? Do deliberate metaphors really exist? If so, are they worth investigating? [8,25]. Nevertheless, neither strand of pure introspection was convincing enough to persuade the other. Therefore, the second round of debate turned to empirical evidence on deliberate metaphors’ validity.

The second round began with a report of Gibbs’s failed experiment [9]. He designed a case study to check whether people really noticed the deliberation and then experienced cross-domain mappings when encountering pragmatic signals of metaphor [9]. However, the experimental results did not confirm the tenet. This failed test posed a serious challenge to the psychological reality of Deliberate Metaphor Theory (p. 87), and Gibbs urged its proponents to provide convincing empirical evidence for its tenet.

In response, Steen indicated the following three inadequacies in the experiment’s conceptualization and operationalization [27]. (1) The tested pragmatic signals either had nothing to do with metaphor signaling or were just intensifiers to draw attention to the expression’s literal meaning (p. 3). That is, the pragmatic signals were not valid for deliberate metaphors. (2) Measurement of the effect of metaphors was problematic (p. 4). Moreover, even for unproblematic measurement, the small number of participants (12 per condition) was too unreliable. (3) The entire experiment tested interpretations of only one linguistic metaphor, and one case could not be “representative of the entire phenomenon” (p. 5). Therefore, Steen did not accept this study as a *failed* test of the Deliberate Metaphor Theory.

As a follow-up, Gibbs [28] commented on two inadequacies that Steen indicated. (1) If the employed pragmatic signals were not valid, what are the true signals used to mark deliberate metaphors? Proponents had to present these signals explicitly and empirically to check their communicative effects on receivers. (2) If the measurement was not applicable in deliberate metaphor testing, what questions could be proposed to check Deliberate Metaphor Theory? Advocates had never offered a referenced paradigm for the experimental design and needed to construct reliable empirical paradigms. Therefore, Deliberate Metaphor Theory had to adopt a “firmer empirical vision” if it were to have a “valid value in metaphor studies” (p. 86).

Although opponents’ views of the experiment as failed and proponents’ refusal to accept the experiment as failed complicated the entire picture, this second round did provide some insights that became guidelines for our experimental design in the present study: (1) Valid instances for deliberate metaphors are needed for experimental design. (2) An experiment should be designed that confirms the essential attribute of deliberate metaphor: whether the addressees pay more attention to deliberate metaphors and display a higher tendency of perspective change. (3) An experiment is expected to include a variety of deliberate metaphors.

Next, the third round of debate began with Gibbs and Chen’s (2017) contradiction of Xu, Zhang, and Wu’s [29] article proposing that deliberate metaphor widened the scope of metaphor studies. Contrarily, Gibbs and Chen [10] claimed that Deliberate Metaphor Theory took researchers back to “a Stone Age” (p. 124). In response, Steen [15] stated 11 basic assumptions to clarify deliberate metaphor’s major concerns; this offered an all-encompassing view of metaphors compatible with Conceptual Metaphor Theory and at the same time added new dimensions to the current paradigm (p. 20). Once again, the two strands did not convince each other. The major issue still remains. Does deliberate metaphor really exist? The psychological reality of deliberate metaphor becomes a decisive factor of its value in the existing paradigm.

In a nutshell, although the three rounds of debates intended to verify the reality of deliberate metaphor and the necessity to research the phenomenon, no consensus has been reached. The essential issue is that no supportive evidence has been offered for the psychological reality of deliberate metaphor, and this makes all the relevant discussion groundless. Thus, the present study attempts to design an experiment to attest to the psychological reality of deliberate metaphors, aiming at furthering the investigation into this phenomenon.

## 3. Research Method

### 3.1. Research Questions

As previously explained, the only “failed” empirical experiment on deliberate metaphor was criticized for inadequate conceptualization and operationalization [9,27]. We thus decided to adhere to the definition of deliberate metaphor in designing our research question. As presented at the end of the Introduction section, we propose three hypotheses regarding the reception process based on the definition and tenets of deliberate metaphor. Taken together, these aspects depict deliberate metaphors’ psychological reality checked via attention, perspective adoption, and respective response time. The three hypotheses can be formulated into two research questions in the following.

**Research** **Question** **1:**
*Compared with non-deliberate metaphors, does the addressee pay more attention to deliberate metaphors’ source domain and spend more time on this process?*


**Research** **Question** **2:**
*Compared with non-deliberate metaphors, does the addressee display a greater tendency to adopt the source domain’s perspective and spend more time on this process?*


### 3.2. Participants

From a university in northwest China, 75 native Chinese speakers participated in this study. Their Chinese scores on the National College Entrance Examination (similar to the SAT in the US or A-levels in the UK) were collected to check whether they were homogeneous in language proficiency. Five outliers were dropped because their scores fell outside the mean score range plus or minus three standard divisions. The final participants were seventy (sixty-one females and nine males) whose ages ranged from 18.2 to 21.7 years (M = 20.3, SD = 0.85). The gender imbalance was accepted because no significant difference was detected between males and females in metaphor processing [30]. The study was approved by the Ethical Committee of Bilingual Cognition and Development Lab at Guangdong University of Foreign Studies. All participants were voluntary and had signed an informed consent form. At the end of the experiment, a small amount of money was made available as a reward.

### 3.3. Materials and Methods

As concluded from the second-round debate, a variety of valid deliberate metaphors should be included in the experimental design. However, only one Chinese article investigated the cognitive function of deliberate metaphors in Chinese news reports about COVID-19 [31], discussing three Chinese deliberate metaphors, which were not adequate for the experimental design. Therefore, we constructed a small corpus of news reports from the websites of the three most representative Chinese media: People’s Daily, Xinhua News Agency, and CCTV (China Central Television) News. These contain different sections, for instance, society, livelihood, technology, entertainment, culture, sports, and tourism, which guarantee the diversity of the corpus. From 1 October to 1 November 2020, from each website section, the headline news was downloaded and merged into one document (90,676 words).

Together with the first author, a doctoral student in cognitive metaphorology was invited to identify both deliberate and non-deliberate metaphors. After identifying all metaphors, we checked our inter-rater reliability using Pearson’s correlation analysis. The results showed a significant level (r = 0.87, *p* < 0.05; r = 0.93, *p* < 0.05), which confirmed the reliability of the metaphors we identified. For deliberateness testing, we selected metaphors both of us recognized. Another doctoral student in cognitive metaphorology was invited to grade the deliberateness of these metaphors on a scale from 1 for “least deliberate” to 5 for “extremely deliberate”. The results indicated two typical types of Chinese deliberate metaphors: novel metaphors and conventional metaphors in quotation marks, as in the two following sentences.

(1)脱贫故事在全国各地都火热轮番上演。
tuōpín gùshì zaì quánguó gèdì dōupoverty alleviation stories in country everywhere whole huŏrè lúnfān shàngyăn.burning hot repeatedly perform.Burning hot poverty alleviation stories are being performed repeatedly everywhere in the whole country.(2)人民的拼搏推动中国号“列车”加速前行。
rénmínde pīngbó tuīdòng zhŏngguóhào lièchē jiāsùpeople’s strive push No. China train acceleratedqiánxíngmove forwardPeople’s strive accelerates the “train” No. China to move forward.

Sentence (1) used “火热 (huŏrè, burning hot)” to describe poverty alleviation stories, which was novel for Chinese people. In this way, the writer indicated his deliberateness in drawing people’s attention to this metaphor. Sentence (2) used “列车 (lièchē, train)” to refer to the country of China, which was conventional for Chinese people. Therefore, the writer added quotation marks to draw people’s attention to this metaphor. We decided to include these two types of Chinese deliberate metaphors in the experiment (termed deliberate novel metaphor and deliberate conventional metaphor, respectively).

The final testing materials included 20 sentences with deliberate novel metaphors, 20 with deliberate conventional metaphors (top sentences in deliberateness testing), 20 with non-deliberate metaphors (bottom sentences in deliberateness testing), and 20 literal expressions as baselines (also selected from the self-constructed corpus). All 80 sentences were edited to a length of 16 words. The source domains were located in the 11th and 12th words in sentences and underlined (as in example sentences (1) and (2)). Similarly, the 11th and 12th words in the literal expressions were also underlined. After that, all sentences were programmed into E-Prime 2.0.

The experiment was conducted in a language laboratory, with each participant using a solo computer. Three experienced experiment assistants were present in case of technical issues. Before the test, participants were familiarized with the rubric and the keyboard operation. During the experiment, the computer first displayed “+” in the center of the screen for 500 ms, followed by a blank screen for 500 ms, and then displayed the 80 sentences randomly for 3000 ms. Participants performed the following two tasks on each sentence: grade their attention to the underlined part on a 1–5 scale (1 for “least attention”; 5 for “close attention”), and judge whether they consciously adopted the perspective of the underlined parts (Y for “yes”; N for “no”). For each sentence, the attention task appeared first without time control. After participants pressed a number to rank their attention to the source domain, the second task also appeared without time control. Participants pressed the corresponding letter according to their adopted understanding perspectives as soon as possible once they had made their decision. When the two tasks for one sentence were completed, the next sentence appeared at the screen’s center (Figure 1 displays the procedure for one sentence). This process was repeated for all sentences. No one could leave the lab before all participants had finished the experiments. Average scores of attention and percentages of the choice “Yes” for each sentence and their response times were, respectively, extracted for further analysis.

### 3.4. Results

Research Question 1 tested whether the addressee paid more attention to the deliberate metaphors’ source domain and spent a longer time on this process. The null hypothesis and alternative hypothesis were:

**H_0_:** 
*Compared with non-deliberate metaphors, addressees pay the same degree of attention to deliberate metaphors’ source domains and spend the same amount of response time on them.*


**H_1_:** 
*Compared with non-deliberate metaphors, addressees pay a higher degree of attention to deliberate metaphors’ source domains and spend more response time on them.*


Average scores of attention for the four types of sentences and their respective response times were calculated. Table 1 displays the results.

Table 1 shows that participants paid more attention to both types of deliberate metaphors (3.18 for deliberate novel and 4.63 for deliberate conventional metaphors) than to non-deliberate metaphors (2.45). However, only deliberate novel metaphors (3714.59) required more response time than non-deliberate metaphors (3554.42). These results partially reject the null hypothesis and support the alternative hypothesis. It seems that both types of deliberate metaphors draw more attention from receivers and that deliberate novel metaphors require more processing time. Paired *t*-tests were conducted to check the significance of these differences (Table 2).

Table 2 shows that participants paid significantly more attention to both deliberate novel metaphors (*t* = 12.46, *p* < 0.01, *d* = 1.00) and deliberate conventional metaphors (*t* = 21.10, *p* < 0.01, *d* = 3.37) than to non-deliberate metaphors. Moreover, the difference between the response time required by deliberate novel metaphors and non-deliberate metaphors also reached a significant level (*t* = 2.15, *p* < 0.05, *d* = 0.14). These results further confirm our findings from Table 1, verifying the psychological reality of Chinese deliberate metaphors. Significantly more attention to deliberate metaphors and more response time provide statistical evidence for our assumption.

We also noticed two unexpected findings. First, deliberate conventional metaphors displayed significantly less response time than non-deliberate metaphors (2865.42 and 3554.42, respectively, *t* = −4.84, *p* < 0.01, *d* = 0.29). This might be caused by the experiment’s specific tasks. Generally, non-deliberate metaphors would be processed (categorized) as non-metaphorical expressions [22,32,33]. When participants were asked to judge their degree of attention to non-deliberate metaphors, they might have realized the metaphorical nature of these metaphors, resulting in a longer processing time. On the contrary, quotation marks around deliberate conventional metaphors would have instantly attracted participants’ attention, resulting in a short response time. The second finding is that deliberate novel metaphors and deliberate conventional metaphors displayed significant differences in both degrees of attention and respective response time. This reminds us of the delicate classification of metaphor types in experimental design and results analysis.

Research Question 2 tested whether the addressee would display a higher tendency to adopt the source domain perspective and spend more time on this process. The null hypothesis and alternative hypothesis were:

**H_0_:** 
*Compared with non-deliberate metaphors, addressees display the same tendency to adopt the source domain perspective and spend the same amount of time on this process.*


**H_1_:** 
*Compared with non-deliberate metaphors, addressees display a higher tendency to adopt the source domain perspective and spend a longer time on this process.*


The average percentages of perspective adoption for the four types of sentences and their respective response times were calculated (Table 3).

Table 3 indicates that both deliberate novel and deliberate conventional metaphors demonstrated higher percentages of adopting the source domain perspective than non-deliberate metaphors. Once again, only deliberate novel metaphors required more response time than non-deliberate metaphors. Taken together, these results partially reject the null hypothesis and support the alternative hypothesis. It seems that both types of Chinese deliberate metaphors lead to more perspective adoption and that deliberate novel metaphors demand more processing time. Paired *t*-tests were conducted to check the significance of these differences (Table 4).

Table 4 shows that both deliberate novel metaphors (*t* = 10.91, *p* < 0.01, *d* = 1.28) and deliberate conventional metaphors (*t* = 17.45, *p* < 0.01, *d* = 3.18) displayed a significantly greater tendency to adopt the source domain perspective than non-deliberate metaphors. This further confirms the finding from Table 3. However, the difference between the response time of deliberate novel metaphors and non-deliberate metaphors did not reach a significant level (*t* = 0.02, *p* > 0.05, *d* = 0.00). This reminds us of the first unexpected finding from Research Question 1. Asking participants to consider their perspective change when processing non-deliberate metaphors might make them rethink the understanding process, resulting in a longer response time. Further experiments are expected to verify this. In addition, significant differences between deliberate novel metaphors and deliberate conventional metaphors in both perspective adoption (*t* = −12.59, *p* < 0.01, *d* = 1.87) and respective response time (*t* =.4.74, *p* < 0.01, *d* = 0.48), also suggest a detailed classification of metaphors in experimental design and results analysis.

### 3.5. General Discussion

Based on the deliberate metaphor’s definition and main tenets, the present study proposed two hypotheses about its processing procedure: In comparison to non-deliberate metaphors, deliberate metaphors would draw more attention from addressees and bring about more perspective change, both of which require more processing time. Our results confirmed these hypotheses except for the longer response time for non-deliberate metaphors. Specifically, participants paid significantly more attention to deliberate metaphors’ source domains and demonstrated greater percentages of adopting these source domains’ perspectives. Compared with non-deliberate metaphors, deliberate novel metaphors cost participants more processing time, while deliberate conventional metaphors cost less time. The task’s possible effect on non-deliberate metaphors’ longer response time will be checked in further studies. Implications of the confirmed hypotheses are listed below.

The significantly greater attention to deliberate metaphors’ source domain and the higher percentage of adopting their understanding perspectives suggest that a distinct mechanism might be involved in processing deliberate metaphors and that deliberate metaphors and non-deliberate metaphors activate different mental processes in receivers’ minds. In other words, they can be classified into different types of metaphors. As such, this offers supportive statistical evidence for the psychological reality of deliberate metaphors and should resolve the debate about their existence.

In the first round, opponents inquired how we could verify receivers’ attention to deliberate metaphors and subsequent cross-domain mappings [8]. However, proponents did not provide convincing evidence, leading to doubt about the existence of deliberate metaphors. Our research questions correspond to this inquiry, and the results provide empirical evidence for receivers’ attention to deliberate metaphors and subsequent perspective adoption. This aligns with the finding that English deliberate metaphors required longer gaze duration [34]. The inquiry has thus received a positive reply.

In the second round, Gibbs [9] failed to verify the receivers’ perspective change with the presence of pragmatic signals. His experiment was criticized for three inadequacies: invalid signals for deliberate metaphors, problematic measurement, and testing of just one metaphor. We designed our present experiment in consideration of these inadequacies and were successful in verifying the receivers’ perspective change. Hopefully, this experiment paradigm can provide a reference for future studies in response to opponents’ requirements in this round of debate [28].

In the third round, two stands turned to the value of deliberate metaphors for general metaphor study [10]. If deliberate metaphors have been evidenced as a distinct metaphor type, they should be worth investigation and research. This echoes the claim that deliberate metaphor extends conceptual metaphors [29,35].

Moreover, deliberate novel metaphors and deliberate conventional metaphors showed significant differences in receivers’ attention, perspective change, and respective response times. This suggests that they involve a distinct processing mechanism. Deliberate novel metaphors resort to novelty to indicate deliberateness in use. Novelty is an intrinsic property of unfamiliar (novel) metaphors [36]. Unfamiliarity could attract greater attention, but at the same time might cause difficulty in processing because receivers need to create new relations between two distinct concepts. For deliberate conventional metaphors, quotation marks could easily draw people’s attention, and the metaphor’s conventionality requires no construction of new relations [33]. These different mechanisms led to divergent results in the present study and remind us to classify metaphors more meticulously to unearth their particular mechanism [37].

This study has its limitations. First, all our findings were based on data from the Chinese language. As a different language typology from English, Chinese may display specific traits in deliberate metaphor processing [38,39]. The different geographical environments, the long-established traditions [39], and the diverse cultural backgrounds [1,40] cause systematic variation in metaphor interpretations. Future investigation should check whether our findings can be generalized to other languages with different geographical, social, and cultural backgrounds. Moreover, with its accurate record of participants’ responses and response times [41], our behavioral data is sufficient for our present purpose. However, time is somewhat of a partial proxy for attention. Therefore, it is suggested that other technologies such as eye-tracking and fNIRS (functional Near-Infrared Spectroscopy) be employed in the future to find more direct evidence for our present findings. What is more, our experimental design was inspired by Steen’s [16] suggestion that the fundamental difference between deliberate and non-deliberate metaphors hinges on attention. Steen proposed that attention to metaphor, the moment where embodied cognition and social interaction meet, is an interesting site for careful investigation. Nevertheless, he and his group did provide a referential research paradigm. Although the present study succeeded in verifying the hypothesized differences, the tasks relied more on conscious awareness. Deliberate metaphors involve unconscious processing as well [12]. We expect to adopt fNIRS to get receivers’ brain images in the areas of the frontal lobe and parietal lobe responsible for attention during natural communication with deliberate and non-deliberate metaphors to check our findings in following investigations.

## 4. Conclusions

The deliberate metaphor was proposed to tackle the paradox in general metaphor studies that not all metaphors are processed in a metaphorical manner. Nevertheless, deliberate metaphor has been under constant debate about its psychological reality since its debut. Although proponents presented a theoretical account of its psychological reality according to certain aspects, they did not provide supportive statistical data to demonstrate that psychological reality. In contrast, although opponents resorted to empirical experiments to evidence its psychological reality, they reported a failed test. Therefore, verification of its psychological reality became an urgent issue for further development of deliberate metaphor research.

In view of this situation, the present study employed a behavioral experiment to test the psychological reality of Chinese deliberate metaphors. In our experiment, 70 native Chinese university students were required to grade their attention to the underlined parts of presented sentences and to judge whether they had adopted the understanding perspectives of these underlined parts. Results confirmed that receivers paid more attention to source domains of deliberate metaphors and were more likely to adopt their understanding perspectives. At the same time, receivers spent more time processing deliberate novel metaphors. These results suggest a distinct mechanism involved in deliberate metaphor processing, thus proving its psychological reality. Whether non-deliberate metaphors’ longer response time compared with deliberate conventional metaphors was caused by our experiment’s specific tasks needs to be re-tested in further studies.

This is the first successful verification of the psychological reality of deliberate metaphors from the reception side. As far as we know, this is one of the few studies adopting an experimental approach to help the research move forward. If it is the case, deliberate metaphors should involve distinct mechanisms from non-deliberate metaphors. In this way, the doubt and debate on the necessity and value to investigate deliberate metaphors will be dispersed. As such, we need to construct a more comprehensive framework with both deliberate and non-deliberate metaphors [29]. This requires collaboration not merely between proponents and opponents of deliberate metaphors, but among all researchers in the field of metaphor study. The issue of importance is to build an integrated research paradigm that accounts for all types of metaphors identified to date.

Future studies should check whether this finding can be generalized to other languages. In addition, deliberate metaphor implies two subjectivities: the producers who intend to use the metaphor and the receivers who attend to the metaphor. As such, the psychological reality of producers also emerges as an important issue to be investigated in future studies. Moreover, if possible, some advanced technologies such as eye-tracking, ERP (Event-related Potential), and fNIRS could be employed to offer more direct evidence from the brain.

## Figures and Tables

**Figure 1 brainsci-13-00160-f001:**
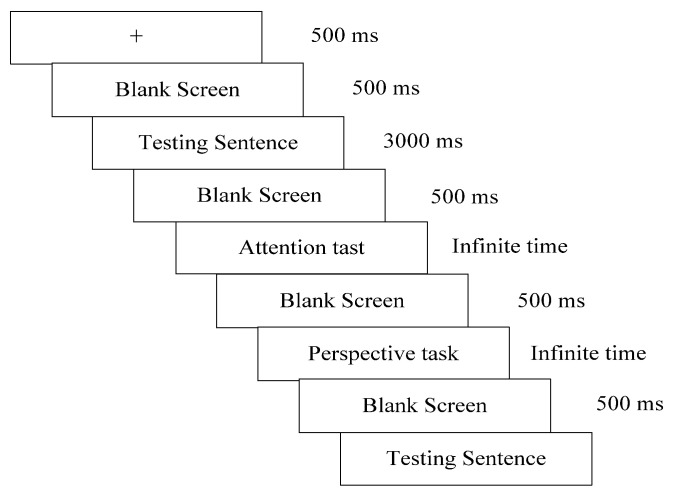
Procedure for one sentence.

**Table 1 brainsci-13-00160-t001:** Average scores of attention and respective response times.

	Attention	Response Time (Ms)
Deliberate novel metaphors	3.18 (SD = 0.66)	3714.59 (SD = 1097.58)
Deliberate conventional metaphors	4.63 (SD = 0.46)	2865.42 (SD = 1114.67)
Non-deliberate metaphors	2.45 (SD = 0.79)	3554.42 (SD = 1201.05)
Literal expressions	1.90 (SD = 0.75)	3150.84 (SD = 1132.98)

**Table 2 brainsci-13-00160-t002:** Paired *t*-tests on the overall scores of attention and response times (CI = 95%).

	Attention				Response Time			
	*t*	df	Sig. (2-Tailed)	*d*	*t*	df	Sig. (2-Tailed)	*d*
DN-ND	12.46	69	0.00	1.00	2.15	69	0.04	0.14
DC-ND	21.10	69	0.00	3.37	−4.84	69	0.00	0.29
DN-LE	17.63	69	0.00	1.81	6.12	69	0.00	0.51
DC-LE	27.16	69	0.00	4.39	−2.10	69	0.04	0.25
DN-DC	−17.99	69	0.00	2.55	6.90	69	0.00	0.77

(DN = deliberate novel metaphors, DC = deliberate conventional metaphors, ND = non-deliberate metaphors, LE = literal expressions).

**Table 3 brainsci-13-00160-t003:** Average percentages of perspective adoption and respective response times.

	Percentage	Response Time (Ms)
Deliberate novel metaphors	53.50 (SD = 20.75)	1381.31 (SD = 859.93)
Deliberate conventional metaphors	89.86 (SD = 18.04)	1008.12 (SD = 677.07)
Non-deliberate metaphors	26.34 (SD = 21.72)	1378.82 (SD = 1156.18)
Literal expressions	8.29 (SD = 14.14)	888.60 (SD = 1444.54)

**Table 4 brainsci-13-00160-t004:** Paired *t*-tests on the overall scores of percentage and response times (CI = 95%).

	Percentage				Response Time			
	*t*	df	Sig. (2-Tailed)	*d*	*t*	df	Sig. (2-Tailed)	*d*
DN-ND	10.91	69	0.00	1.28	0.02	69	0.98	0.00
DC-ND	17.45	69	0.00	3.18	−3.32	69	0.00	0.39
DN-LE	16.09	69	0.00	2.55	3.17	69	0.00	0.41
DC-LE	24.32	69	0.00	5.03	0.80	69	0.43	0.11
DN-DC	−12.59	69	0.00	1.87	4.74	69	0.00	0.48

## Data Availability

Data is available upon request of editors and reviewers.

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
