# Peer review of "The Psychological Reality of Chinese Deliberate Metaphors from the Reception Side: An Experimental Approach"

_brainsci, 2023, doi:10.3390/brainsci13020160_

Round 1

Reviewer 1 Report

This is an important piece to the study of metaphor generally and specifically to the more applied question of the existence, function, and utility of deliberate metaphor. The paper is very clearly written and well-referenced.  But would encourage the authors to consider the following comments:

1. I would love to see some discussion of the more applied value of deliberative metaphor (for example as a tool for science translation or communication of complex phenomena). To me, deliberative metaphor is so interesting because of its implications for applied communications. The paper would be stronger with some of this discussion included. 

2. It would be great to define "pragmatic signals" where this language first appears --this would improve the readability of the paper. 

3. On page 3 line 130-131: it is unclear what is meant by the "essence of deliberate metaphors". This could use some rewriting just to make things a bit sharper and more precise. 

4. I found the section at the top of page 4 very helpful--where the features of deliberate metaphor that are being tested/ interrogated are clearly spelled out. This was my biggest critique up to this point (that is was unclear how the dimensions being tested proved/tested the reality of deliberate metaphor). It is very clearly spelled out here but I would like to have some of this clarity earlier on the paper. If the authors can move it up it will preventatively answer some fundamental questions/critiques. 

5. I think it needs to be noted in the limitations that time is somewhat of an imperfect proxy for attention--partial at least. Response time as a way to measure attention has its problems and this needs to be noted. 

6. My biggest issue is around methodology. This does not mean that the paper isn't of value or that it shouldn't be published but that these limitations must be much more strongly and clearly presented as such. The concern is around using respondents' on evaluations of their attention to source domain and even more problematically whether using self-assessment to measure whether they applied to source domain in their thinking. Much of this processing works under the level of conscious awareness--even in deliberate metaphors--and as such is much better measured in task that recruit and measure the function rather than asking respondents to do so. To me this is a very significant limitation and must be more clearly noted as such. I would also like to see some discussion about how this limitation might be managed in future work (other than eye-tracking). Metaphors and their processing can still be deliberate without people being able to assess their own processing their of--and their own assessments of attention and application seem inherently open to error given what we know about metaphor processing--even with deliberate metaphors. 

All in all a very good paper and I very much enjoyed reading it. Very interesting and important work--thank you to the authors for a great piece of work and contribution to the field! 

Author Response

Dear Editors and Reviewers,

Many thanks for your valuable suggestions on our manuscript! Accordingly, we provided the required information, clarified some unclear concepts, and improved the presentation of the methodology. The details of revision will be presented in the order of the given comments.

Reviewer 1:

This is an important piece to the study of metaphor generally and specifically to the more applied question of the existence, function, and utility of deliberate metaphor. The paper is very clearly written and well-referenced.  But would encourage the authors to consider the following comments:

  1. I would love to see some discussion of the more applied value of deliberative metaphor (for example as a tool for science translation or communication of complex phenomena). To me, deliberative metaphor is so interesting because of its implications for applied communications. The paper would be stronger with some of this discussion included. 

Response: We have now added discussion on the communication dimension of deliberate metaphor in the revision.

The following is the added part in the revised version (line 41 on p.1):

Accordingly, deliberate metaphor adds a third dimension to metaphor study, the communication dimension [27]. It invites addressees to view the topic from the perspective of the source domain. This function makes it a valuable tool in communicating knowledge in academic discourse, science communication and other communication of complex phenomenon [1]. In these communications, addressers deliberately use metaphors to inform or explain some difficult concepts or topics, which finally change addressees’ perspectives [27].

  1. It would be great to define "pragmatic signals" where this language first appears --this would improve the readability of the paper. 

Response: We have now added explanation for “pragmatic signals” in the revision.

The following is the added part in the revised version ( line 54 on p.2):

pragmatic signals (also called tuning devices, including certain discourse markers, intensifiers, and comparatives and the like, such as ‘well’, ‘quite’, and ‘like’)

  1. On page 3 line 130-131: it is unclear what is meant by the "essence of deliberate metaphors". This could use some rewriting just to make things a bit sharper and more precise. 

Response: We have now rephrased the expression and also added explanation for it.

The following is the added part in the revised version ( line 150 on p.4):

An experiment should be designed that confirms the essential attribute of deliberate metaphor: whether the addressees pay more attention to deliberate metaphors and display a higher tendency of perspective change.

  1. I found the section at the top of page 4 very helpful--where the features of deliberate metaphor that are being tested/ interrogated are clearly spelled out. This was my biggest critique up to this point (that is was unclear how the dimensions being tested proved/tested the reality of deliberate metaphor). It is very clearly spelled out here but I would like to have some of this clarity earlier on the paper. If the authors can move it up it will preventatively answer some fundamental questions/critiques. 

Response: Sorry for the confusion! We have now moved this part to line 60 to 72 on page 2.

  1. I think it needs to be noted in the limitations that time is somewhat of an imperfect proxy for attention--partial at least. Response time as a way to measure attention has its problems and this needs to be noted. 

Response: We have now added this point to the limitations part on page 10 line 408.  

  1. My biggest issue is around methodology. This does not mean that the paper isn't of value or that it shouldn't be published but that these limitations must be much more strongly and clearly presented as such. The concern is around using respondents' on evaluations of their attention to source domain and even more problematically whether using self-assessment to measure whether they applied to source domain in their thinking. Much of this processing works under the level of conscious awareness--even in deliberate metaphors--and as such is much better measured in task that recruit and measure the function rather than asking respondents to do so. To me this is a very significant limitation and must be more clearly noted as such. I would also like to see some discussion about how this limitation might be managed in future work (other than eye-tracking). Metaphors and their processing can still be deliberate without people being able to assess their own processing their of--and their own assessments of attention and application seem inherently open to error given what we know about metaphor processing--even with deliberate metaphors. 

Response: Thank you very much for your enlightening comment! We have now added the reason why we designed these tasks, and also provided possible solutions in the future study.

The following is the added part in the revised version ( line 408 on p.10):

However, time is somewhat of an partial proxy for attention. Therefore, it is suggested that other technologies such as eye-tracking and fNIRS be employed in the future to find more direct evidence for our present findings. What’s more, our experiment design was inspired by Steen’s [34] suggestion that the fundamental difference between deliberate and non-deliberate metaphors hinge on attention. Steen proposed that attention to metaphor, the moment where embodied cognition and social interaction meet, is an interesting site for careful investigation. Nevertheless, he and his group did provide a referential research paradigm. Although the present study succeeded in verifying the hypothesized differences, the tasks relied more on conscious awareness. Deliberate metaphors involve unconscious processing as well [29]. We expect to adopt fNIRS to get receivers’ brain images in the areas of the frontal lobe and parietal lobe responsible for attention during natural communication with deliberate and non-deliberate metaphors to check our findings in following investigation.

Thank you again for all of your valuable comments!

Reviewer 2 Report

This is an interesting study which was designed well. A few areas to consider:

·        In the Abstract the authors talk about a hypothesis but in the Methodology they only outline research questions. This has to be clarified and the relevant phrasings have to be fixed because hypotheses and research questions are not the same.

·        What is not clear is what the contribution of the study to scholarship is. Although, the authors outline the contribution in the Discussion, it seems that it needs more in-depth insights.

·        Has the use of deliberate and non-deliberate metaphors been affected by culture? In which ways? What the literature shows? Has the study touched on this question? The authors talk about the importance of language in the limitations section, but it is not really language itself but the meanings and emotional load words and sounds carry because of the use of language in culture over time.

Author Response

Dear Editors and Reviewers,

Many thanks for your valuable suggestions on our manuscript! Accordingly, we provided the required information, clarified some unclear concepts, and improved the presentation of the methodology. The details of revision will be presented in the order of the given comments.

Reviewer 2:

This is an interesting study which was designed well. A few areas to consider:

  1. In the Abstract the authors talk about a hypothesis but in the Methodology they only outline research questions. This has to be clarified and the relevant phrasings have to be fixed because hypotheses and research questions are not the same.

Response: Sorry for this confusion. We have firstly put forward hypothesis, and formulated research questions based on these hypothesis. We have now made this clear in the revision.

The following is the added part in the revised version ( line 176 on p.4):

As presented at the end of the Introduction section, we propose three hypotheses regarding the reception process based on the definition and tenets of deliberate metaphor. Taken together, these aspects depict deliberate metaphors’ psychological reality checked via attention, perspective adoption, and respective response time. The three hypotheses can be formulated into two research questions in the following.

  1. What is not clear is what the contribution of the study to scholarship is. Although, the authors outline the contribution in the Discussion, it seems that it needs more in-depth insights.

Response: Thank you for this suggestion. We have now added more discussion on the contribution.

The following is the added part in the revised version ( line 443 on p.10):

This is the first successful verification of the psychological reality of deliberate metaphors from the reception side. As far as we know, this is one of the few studies adopting an experimental approach to help the research move forward. If it is the case, deliberate metaphors should involve distinct mechanism from non-deliberate metaphors. In this way, the doubt and debate on the necessity and value to investigate deliberate metaphors will be dispersed. As such, we need to construct a more comprehensive framework with both deliberate and non-deliberate metaphors [39]. This requires collaboration not merely between proponents and opponents of deliberate metaphors, but among all researchers in the field of metaphor study. The issue of importance is to build an integrated research paradigm that accounts for all types of metaphors identified to date.

  1. Has the use of deliberate and non-deliberate metaphors been affected by culture? In which ways? What the literature shows? Has the study touched on this question? The authors talk about the importance of language in the limitations section, but it is not really language itself but the meanings and emotional load words and sounds carry because of the use of language in culture over time.

Response: Yes, social and cultural backgrounds are also sources for variation in metaphor interpretation. We have now added discussion on this point in the revision.

The following is the added part in the revised version ( line 401 on p.9):

As a different language typology from English, Chinese may display specific traits in deliberate metaphor processing [37-38]. The different geographical environment, the long established tradition [38] and the diverse cultural backgrounds [18, 25] cause systematic variation in metaphor interpretations. Future investigation should check whether our findings can be generalized to other languages with different geographical, social and cultural backgrounds.

Thank you again for all of your valuable comments!
